# Impacts of maritime shipping on air pollution along the U.S. East Coast

Maryam Golbazi[1] and Cristina Archer[1]

[1]Center for Research in Wind (CReW), University of Delaware, 221 Academy Street, Newark, DE 19716, USA

**Correspondence:** Maryam Golbazi, Center for Research in Wind (CReW), University of Delaware, 221 Academy Street, Newark, DE 19716, USA (mgolbazi@udel.edu)

**Abstract.**

Air pollution is considered a leading threat to human health in the U.S. and worldwide. An important source of air pollution in coastal areas is the globally increasing maritime shipping traffic. In this study, we take a high-resolution modeling approach to investigate the impacts of ship emissions on concentrations of various atmospheric pollutants, under the meteorological conditions and emissions of the year 2018. We utilize the Comprehensive Air Quality Model with extensions (CAMx) to simulate transport, diffusion, and chemical reactions, and the Weather Research and Forecasting (WRF) model to provide the meteorological inputs. We focus on four criteria pollutants – fine particulate matter with a diameter smaller than $2.5\,\mu m$ ($PM_{2.5}$), nitrogen dioxide ($NO_2$), sulfur dioxide ($SO_2$), and ozone ($O_3$) – as well as nitrogen oxide (NO), and we calculate their concentrations in the presence and absence of the ship emissions along the U.S. East Coast, particularly in the proximity of major ports.

We find that ship emissions increase the $PM_{2.5}$ concentrations over the ocean and sparse areas inland. The 98-th percentile of the 24-hour average $PM_{2.5}$ concentrations (the "design value" used by the U.S. Environmental Protection Agency) increased by up to $3.2\,\mu g\,m^{-3}$ in some coastal areas. In addition, ships contribute significantly to $SO_2$ concentrations, up to 95% over the Atlantic and up to 90% over land in coastal states, which represents a ~45 ppb increase in the $SO_2$ design values in some states. The 98-th percentile of the hourly $NO_2$ concentrations also increased by up to 15 ppb at the major ports and along the shore. In addition, we find that the impact of shipping emissions on $O_3$ concentrations is not uniform, meaning that ships affect ozone pollution in both positive and negative ways. Over the ocean, $O_3$ concentrations were significantly higher in the presence of ships, but in major coastal cities $O_3$ concentrations decreased in the presence of ships. Our simulation results show that ships emit significant amounts of fresh NO in the atmosphere, which then helps scavenge $O_3$ in VOC-limited areas, such as major ports. By contrast, over the ocean ($NO_x$-limited regime), enhanced $NO_x$ concentrations due to ships contribute to the formation of $O_3$ and therefore enhance $O_3$ concentrations. Overall, due to the dominant southwesterly wind direction in the region, the impacts of ships on air pollutants mainly remain offshore. However, in coastal states near major ports, the impacts are significantly important.

# 1 Introduction

Globally, it is estimated that, in 2019, ambient air pollution, particularly particulate matter (PM) and ozone ($O_3$), was responsible for 4.5 million premature deaths worldwide (Fuller et al., 2022). This ranked air pollution as a leading risk factor in the Global Burden of Disease Study by the Institute for Health Metrics and Evaluation in 2019 (Murray and Lopez, 1996). Meanwhile, ship traffic is globally increasing and is becoming an important source of air pollution, especially in coastal areas (Corbett and Fischbeck, 1997; Eyring et al., 2010b; Schnurr and Walker, 2019). Sea transport accounts for 80% of goods transported worldwide (Schnurr and Walker, 2019), while recent studies estimate demand growth of almost 40% for seaborne trade by 2050 (Serra and Fancello, 2020). Marine vessels are important sources of air pollutants, emitting sulfur oxides ($SO_x$), nitrogen oxides ($NO_x = NO + NO_2$), particulate matter (PM), carbon monoxide (CO), volatile organic compounds (VOC), and carbon dioxide ($CO_2$) (Corbett et al., 2007; Eyring et al., 2007a; Eyring, 2008; Smith et al., 2015). Low-grade marine fuel oil contains 3,500 times more sulfur than road diesel (Wan et al., 2016). Studies report fuel consumption of ocean-going ships between 200-290 million metric tons for the year 2000 (Corbett and Köhler, 2003; Endresen et al., 2007). Ships are responsible for about 15% of all global anthropogenic $NO_x$ emissions and 4–9% of sulfur dioxide $SO_2$ emissions. In addition, oceangoing ships are estimated to emit 1.2–1.6 million metric tons (Tg) of PM annually (Corbett et al., 2007; Eyring et al., 2010b; Viana et al., 2014).

About 70% of ship emissions occur within 400 km of the shore (Corbett et al., 1999; Eyring et al., 2005; Endresen et al., 2003). Thus, ships can be a major source of pollution in coastal areas and can impact human health. For instance, ship emissions in East Asia have caused 14,500–37,500 premature deaths in 2013, the amount of which had doubled since 2005 (Liu et al., 2016). Similarly, particulates emitted from ships cause 60,000 cardiopulmonary and lung cancer deaths each year worldwide (Corbett et al., 2007). Studies from different parts of the world like China show that shipping emissions increased the annual averaged $PM_{2.5}$ concentrations in the eastern coastal regions up to $5.2\,\mu g\,m^{-3}$, which was carried 900 km inland (Lv et al., 2018). In Europe, although the increase in $PM_{2.5}$ concentrations by ships is found to be small, their relative contribution is large because of the low background $PM_{2.5}$ concentrations (Viana et al., 2009; Aksoyoglu et al., 2016).

Fine particulate matter ($PM_{2.5}$) is a harmful air pollutant that consists of microscopic particles with a diameter smaller than $2.5\,\mu m$. These particles can penetrate human lungs and even the bloodstream and cause serious health problems (U.S. Environmental Protection Agency (EPA), 2020b). $NO_x$ are a group of highly reactive gases; although seven compounds are technically part of the $NO_x$ family (NO, $NO_2$, nitrous oxide $N_2O$, dinitrogen dioxide $N_2O_2$, dinitrogen trioxide $N_2O_3$, dinitrogen tetroxide $N_2O_4$, dinitrogen pentoxide $N_2O_5$), the most abundant are NO and $NO_2$, but only $NO_2$ is actually regulated in the U.S. $NO_2$ is harmful to humans by irritating the human respiratory system and to the environment by creating acid rain (U.S. Environmental Protection Agency (EPA), 2020a; Lin and McElroy, 2011); it is also a precursor to tropospheric ozone ($O_3$) formation, which has further negative impacts on human health (, EPA). Similarly, short-term exposure to $SO_2$ can harm the human respiratory system. These four pollutants – $PM_{2.5}$, $SO_2$, $NO_2$, and $O_3$ – are both primary (i.e., they can be directly emitted into the atmosphere) and secondary (i.e., they can also form after chemical reactions in the atmosphere) pollutants. Here, we will focus on these four pollutants which are among the seven "criteria" pollutants that are regulated at the federal level by the

U.S. Environmental Protection Agency (EPA) via the National Ambient Air Quality Standards (NAAQS)(U.S. Environmental Protection Agency (EPA), 2022b).

Due to the complex nature of the atmosphere and its processes, such as chemical reactions, transport, and diffusion, high concentrations of these pollutants are not necessarily found where their emissions are highest. Therefore, although the ship emissions are released in marine environments, the atmospheric conditions can play an essential role in transporting those pollutants, some of which are precursors for the formation of secondary EPA-regulated pollutants, like $O_3$.

Ozone pollution is one of the main focuses of this study. The rate of ozone production can be limited by the concentration of either VOC or $NO_x$ and depends on the relative sources of hydroxyl radical (OH) and $NO_x$ (Finlayson-Pitts and Pitts Jr, 1993). When the rate of OH production is greater than the rate of $NO_x$ production, the rate of ozone production is $NO_x$-limited. In this situation, ozone concentrations are sensitive to $NO_x$ emissions rather than VOC concentrations. In contrast, when the rate of OH production is less than the rate of $NO_x$ production, ozone production is VOC-limited. In this case, ozone is most effectively reduced by lowering VOC concentrations. NOx is generally higher where human mobility and transportation are higher (Archer et al., 2020) and While $O_3$ is generally $NO_x$-limited in rural areas and downwind suburban areas, in urban areas with high population and high traffic emissions $O_3$ is often VOC-limited (Seinfeld and Pandis, 1998). Motor vehicles are among the major sources of ozone pollution in the region through their $NO_x$ and VOC emissions (Niemeier et al., 2006; Yao et al., 2015; Zhang et al., 2014). However, the impact of ocean ship emissions along the East Coast of the United States is lacking in the literature and we fill this gap in this study. In locations that exceed the EPA ozone standards by only 2-3 ppb, like the small state of Delaware (Moghani et al., 2018), the ship contribution could be of even higher importance.

Here, we explore the impacts of ocean-going ship emissions on the air quality along the U.S. East Coast by utilizing the Comprehensive Air Quality Model with extensions (CAMx) for our simulations. For the first time, we use the most recent high spatial (4 km) and temporal (hourly) resolution ship emission data from the EPA's National Emission Inventory (NEI). We also include the ship stack height to consider at what vertical layer the emissions are emitted into the atmosphere, to be able to account for stability and atmospheric impacts on the pollutants. We investigate the pollution concentrations in a control scenario based on the shipping emissions in the year 2018. Then, we conduct another simulation for a hypothetical condition where we eliminate the ship emissions altogether while keeping everything else the same. The difference between the two scenarios gives insights into the net contribution of the ships to air pollution.

Seasonal variations in the impact of shipping on various pollutants have been documented in prior studies. For example, Eyring et al. (2010) noted that during Mediterranean summer conditions, characterized by slow atmospheric transport, strong solar radiation, and limited washout, primary ship emissions accumulate, and secondary pollutants form. They reported that secondary sulfate aerosols from shipping were responsible for 54% of the average sulfate aerosol concentration in the region during the summer. Our findings along the US East Coast align with these results, highlighting the substantial contribution of ships to $SO_2$ pollution during the summer season. Furthermore, they observed that in winter, shipping NOx emissions could lead to ozone depletion in northern Europe (Eyring et al., 2010a). In a separate study, Eyring et al., (2007) noted significant variations in simulated $O_3$ levels between January and July, despite a consistent ship emission inventory throughout the year. They found that during winter, additional NOx emissions from shipping led to $O_3$ reduction due to titration, while in summer,

these emissions resulted in relatively modest but positive $O_3$ concentration changes in regions with sufficient solar radiation. They also show that the highest ship impacts on $O_3$ due to the ship emissions were found in July and April, whereas in October and January, the impacts were smaller (Eyring et al., 2007b). In this study, however, we base our analysis on the summer (June 1st – August 31st) when the highest $O_3$ episodes occur.

## 2 Methods

### 2.1 Setup of the WRF-CAMx modeling system

We take a modeling approach to explore the pollution concentration across the study domain. The models used in this study are the Weather Research and Forecasting (WRF) model, version 4.3, and the Comprehensive Air quality Model with extensions (CAMx) version 7.1 with the Carbon Bond version 6 revision 5 (CB6r5) chemical mechanism. WRF is developed at the National Center for Atmospheric Research (NCAR) (Skamarock et al., 2019) and is one of the most widely used numerical weather prediction models. CAMx is a modular, Eulerian, 3-dimensional photochemical air quality model (Ramboll Environment and Health, 2020) that simulates the emission, production, advection, diffusion, chemical transformation, and removal of atmospheric pollutants at regional scales and is among the few that are recommended by the EPA for regulatory purposes (U.S. Environmental Protection Agency (EPA), 2022a). We use the WRF-CAMx modeling system to conduct simulations of two separate scenarios, based on the exact same setup and inputs: the first scenario includes the ship emissions (WithShips) while in the second hypothetical scenario we remove the ship emissions altogether (NoShips). The difference in pollution levels between the two cases provides the net contribution of ship emissions to regional air quality.

CAMx requires input data to characterize meteorology and chemistry, initial and boundary conditions for all the modeling domains, and other environmental conditions such as the photolysis rates. Meteorology is an essential factor in the formation of many secondary pollutants, both directly and indirectly. Atmospheric stability plays a significant role in determining pollutant faith (Arya et al., 1999). In CAMx, the plume rise calculations for point sources including the CMV emissions depend on meteorological conditions and atmospheric stability to determine what vertical layer the emissions are emitted in. In the summertime, various atmospheric stabilities have been found to be dominant over the Atlantic Ocean depending on different locations (Golbazi et al., 2022; Golbazi and Archer, 2019; Archer et al., 2016). We use the WRF model to provide meteorological inputs to CAMx. The publicly available WRF-CAMx data processing program (Ramboll Environment and Health, 2020) is used to generate CAMx meteorological input files from WRF output files. Details on the WRF model setup are provided in Table 1. Our period of study is the summer of 2018, selected to reflect the most recent emission inventory available (discussed next in Section 2.2). Photolysis rate inputs to CAMx were calculated using the Tropospheric Ultraviolet and Visible (TUV) radiative transfer and photolysis model (https://www2.acom.ucar.edu/modeling/tropospheric-ultraviolet-and-visible-tuv-radiation-model).

**Table 1.** Details of the WRF-CAMx model setup.

| | |
|---|---|
| Simulation period | 1 June – 30 August, 2018 |
| Horizontal grid resolution | 4 km |
| Vertical layers | 35 |
| Lowest model level | 3.5 m AMSL |
| Spin-up time | 48 hours |
| WRF version 4.3 | |
| Initial/boundary conditions | NAM reanalysis, 6-hourly, 12-km resolution |
| LSM | Noah-modified 21-category IGBP-MODIS |
| PBL Scheme | MYNN2 |
| Shortwave radiation | RRTMG shortwave |
| Longtwave radiation | RRTMG scheme |
| SST update | NASA-JPL 1km resolution data |
| Grid size | $400 \times 400$ grid cells |
| CAMx version 7.1 | |
| Chemistry | Carbon bond 6 revision 5 |
| Meteorological inputs | WRF model v4.3 |
| Emission data | EPA/NEI 2018 |
| CMV emissions | Inline point sources |
| Initial/boundary conditions | EPA 2018 |
| Grid size | $315 \times 300$ grid cells |

The domain of this study covers the East Coast of the United States (Figure 1) and includes major cities and highly populated regions. Furthermore, it contains several major ports, which are found to experience high shipping traffic. The meteorological files have $400 \times 400$ horizontal grid points covering the entire CAMx domain, which consists of $315 \times 300$ grid points, the same as the emission files. We impose 35 vertical levels that are closely spaced near the surface and then gradually expand. The top hydrostatic pressure is 20 hPa and the lowest model level is at approximately 3.5 m above mean sea level (AMSL). Details about the model configuration are discussed in Table 1. Both the WRF and CAMx models have a 4-km horizontal resolution, the same as the emission inventory, in order to avoid spatial interpolation of gridded emissions data. To minimize the impacts of the initial conditions on modeling results, we consider at least 48 hours of spin-up time for both models. Furthermore, as the areas of interest are far from the boundaries, the effects of boundary conditions on modeling results are expected to be minimal.

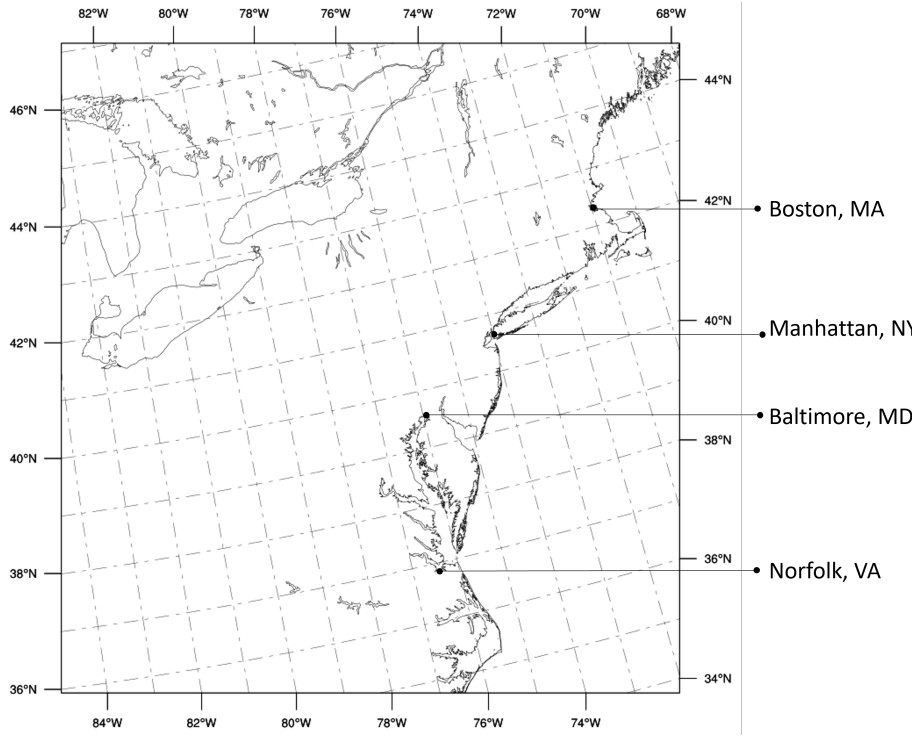

**Figure 1.** The study domain with 315 × 300 grid cells and 4-km horizontal grid resolution.

## 2.2 Emission data

For emission inputs, we use the most recent emission inventory (LISTOS) developed by the EPA, which includes the period May $1^{st}$, 2018 to October $1^{st}$, 2018 (U.S. Environmental Protection Agency (EPA), 2017) at 4-km horizontal grid resolution and hourly temporal resolution. The emissions are distributed on a 315 × 300 grid, which covers the entire East Coast of the U.S. (Figure 1), with 35 layers vertically. Emissions are treated in two basic ways within CAMx: gridded 2D emissions that are released into each grid cell of the modeling domain near the surface (i.e., "area sources", such as traffic or residential heating) and stack-specific "point sources", where each stack is assigned unique coordinates and parameters (i.e., smokestacks or ship chimneys). For inline point source emissions, CAMx computes the plume rise using stack parameters and the hourly emissions for each emission sector.

The 2018 NEI data is based on the year 2017 activity. It contains merged gridded 2D surface emissions, meaning that they are provided as one set of surface emissions that include all the existing 2D emission sectors such as all anthropogenic emissions, aircraft emissions, on-road and non-road emissions, railroad emissions, and agricultural emissions. It also includes biogenic emissions. The 2018 inventory lacks the wildfire emissions for this time and domain. However, our investigation through the wildfire history shows that 2018 was a year with a low number of wildfires especially along the East Coast (https://www.nifc.gov/fire-information/statistics/wildfires) and therefore we do not believe this to significantly impact our

findings. Nonetheless, in future studies, including wildfire emissions upon availability is recommended. In contrast to the 2D grided emissions, the elevated point sources in this inventory are provided for each sector, separately.

For the ship emissions, we use the emission data for the Commercial Marine Vessels (CMV) sector, which includes Category 1, 2 (small engine), and 3 (large engine) ships. These emissions are calculated based on the ship's fuel consumption, ship engine type, ship activity, and emission factors specific to those characteristics. EPA's CMV estimates are computed using 150 detailed satellite-based automatic identification system (AIS) activity data from the US Coast Guard ((U.S. Environmental Protection Agency, 2021, 2020)). Other point sources present in this inventory include electric generation units, point oil, and gas sources, and any other point sources. CAMx computes the time-varying buoyant plume rise using stack parameters and the hourly emissions for each emissions sector, including CMV. Unlike previous EPA data sets, the CMV emissions in 2018 are at a one-hour temporal resolution, which is very important and makes this study the first to utilize hourly emissions for the ships. 155 The initial and boundary conditions for this study are also provided by the EPA and are products of the GEOS-Chem model.

The spatial distribution of the 2D gridded merged anthropogenic emissions are illustrated in Figure 2. It's important to note that $O_3$ is a secondary pollutant, meaning it isn't directly emitted into the atmosphere. Conversely, $PM_{2.5}$ is either a primary or secondary pollutant. Hence, we have specifically generated gridded emission maps for $NO_2$ and $SO_2$, only. The distribution of $NO_2$ emissions closely mirrors the pattern of major highways and roads, as transportation stands out as one of the most 160 significant sources of nitrogen oxides (NOx) emissions. The objective of this figure is to explain the spatial distribution of gridded anthropogenic emissions, shedding light on how concentrations change (Figures 6a and 7a ) in relation to their emission sources.

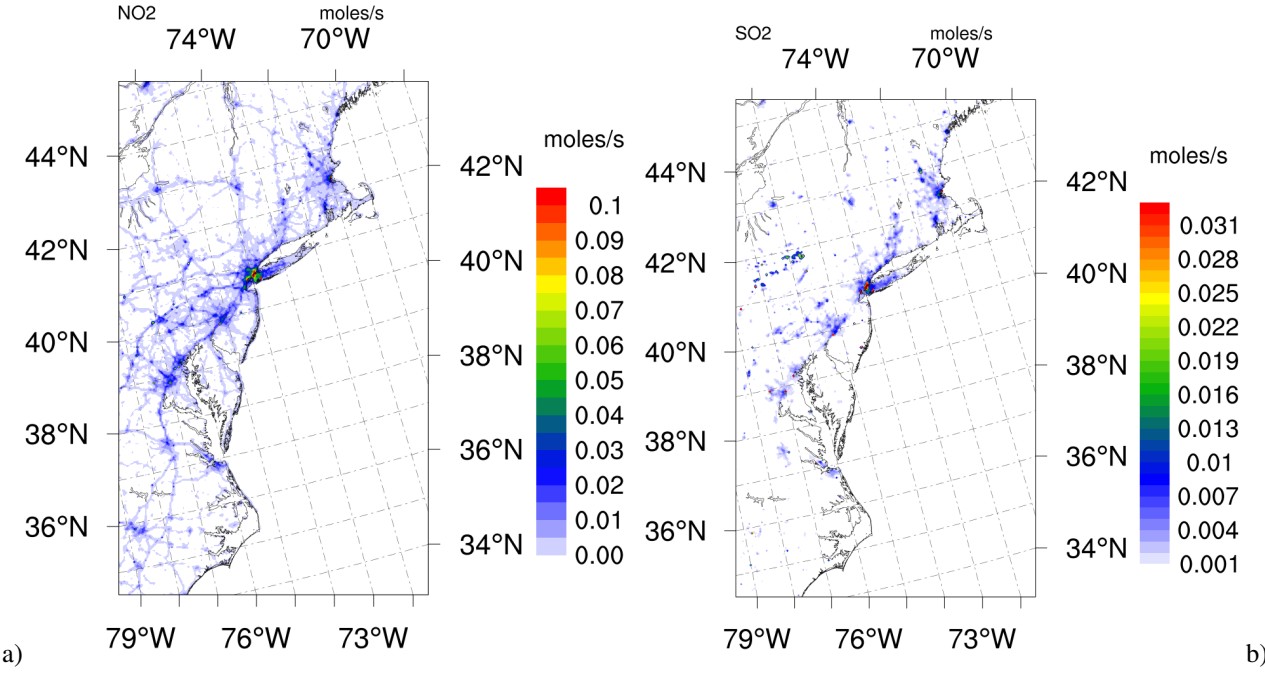

**Figure 2.** Gridded 2D emission distribution across the domain (averaged over time) in moles/s for a) NO$_2$, and b) SO$_2$. The gridded emissions include all the 2D anthropogenic and biogenic emissions and exclude the elevated point sources.

## 3   CAMx model performance analysis

The primary goal of this study is to explore changes to pollution levels between the two examined case studies, one involving the presence of ships and the other without. Despite the instances where CAMx may either under or overestimate pollutant concentrations, it is noteworthy that the model bias remains the same in both scenarios. Consequently, we hold the view that these outcomes are unlikely to have a significant influence on our analysis. Nevertheless, we have thoroughly evaluated the model's performance to maintain transparency in our findings. It's important to acknowledge that uncertainties in air quality modeling can arise from various sources, such as uncertainties in emission inventories ((Foley et al., 2015)), the accuracy of meteorological inputs (Kumar et al., 2019; Ryu et al., 2018; Zhang et al., 2007), numerical noise inherent in the model (Ancell et al., 2018; Golbazi et al., 2022), and numerical approximations.

For our evaluation process of these four pollutants, we rely on measurement data sourced from the Environmental Protection Agency (EPA) AirNow program, which is publicly accessible (https://aqs.epa.gov/aqsweb/documents/data_api.html ). Within the geographical scope of our study, we have access to data from a network of monitoring stations. Specifically, there are a total of 196 stations providing data for O3, and 87, 73, and 118 stations supplying data for SO$_2$, NO$_2$, and PM$_{2.5}$, respectively. This extensive dataset forms the basis of our assessment, enabling us to comprehensively evaluate the CAMx model's performance in replicating real-world air quality conditions for these pollutants. It is worth mentioning that evaluating PM$_{2.5}$ presents

challenges due to the nature of EPA-reported $PM_{2.5}$ measurements in the AirNow database. These values are directly obtained through instrumental measurements, classifying any particle smaller than 2.5 micrograms as a $PM_{2.5}$ species. This method doesn't provide a clear means of distinguishing between the various particles detected by these instruments. In contrast, the $PM_{2.5}$ species in our study are defined based on CAMx model documentation (Ramboll Environment and Health, 2020). This divergence in approach makes a comprehensive $PM_{2.5}$ evaluation challenging and pursuing alternative assessment methods falls beyond the scope of our current study.

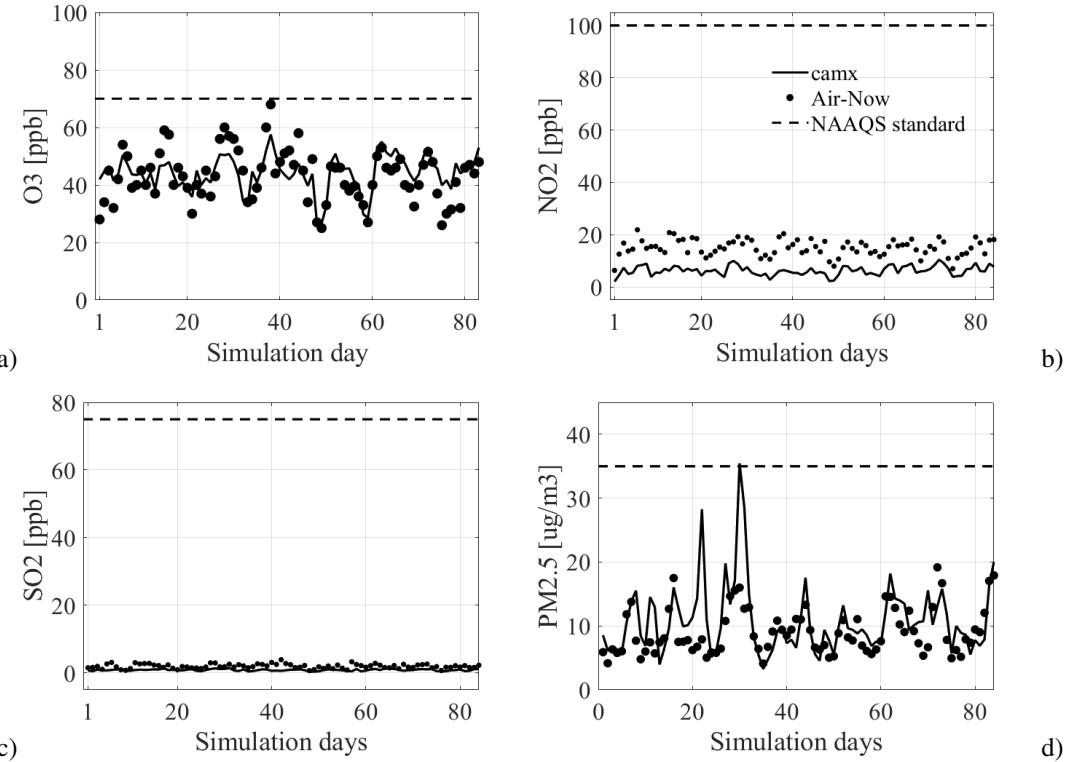

**Figure 3.** Model bias time series; CAMx model performance evaluated against the AirNow measurements; MBE is calculated across all stations at each day a) $O_3$ [ppb], b) $NO_2$ [ppb], and c) $SO_2$ [ppb], and d) $PM_{2.5}$ $[\mu g/m^3]$

Figure 3 illustrates the time series of the AirNow measurements across the simulation days (in black circles), as well as the co-located CAMx outputs for the pollutant of interest in the solid black line. The co-located data are such that they are extracted at the same hour as observations and at the mass point of the grid cell that contains that specific station. Figure 4, on the other hand, illustrates the mean bias error (MBE) calculated at every station and depicts a spatial distribution of the model MBE for each pollutant using the co-located data. CAMx demonstrates a tendency to slightly under- or over-estimate $O_3$ concentrations closer to the coast, and away from the coast, respectively (Figure 4a). Our focus is mainly on locations closer to the coast since that is where we detect the highest impact of shipping emissions. For $O_3$, a calculated MBE of -1.12 ppb indicates a systematic underestimation of around 2.5% across all monitoring stations within the designated domain. Overall, the model effectively

captures the O$_3$ trend and demonstrates a satisfactory level of agreement with observational data, as illustrated in Figure 3a. In addition, CAMx showcases a strong alignment with observational data in terms of SO$_2$ simulations with minimal deviation from the observations.

For PM$_{2.5}$, the model typically underestimates high PM$_{2.5}$ episodes, as is commonly observed in prior studies (Delle Monache et al., 2020; Golbazi et al., 2023). Nonetheless, for the remainder of the time, it demonstrates a strong alignment with observed data, as shown in Figure 3d. Figure 2d reveals that the model bias for PM$_{2.5}$ consistently remains below 5 $\mu$g/m3 for the majority of coastal stations, with only a few exceptions.

Shifting focus to NO$_2$, the model systematically underestimates NO$_2$ concentrations (Figure 3b, and Figure 4b). This observation aligns with findings reported in existing literature (Ma et al., 2006). The notable underestimation of NO$_2$ levels within the model can be attributed to the fact that the monitoring stations are typically situated in close proximity to major roadways characterized by heavy traffic flow, resulting in elevated NO$_2$ emissions. Conversely, NO$_2$ concentrations at locations farther away from these monitoring stations tend to be significantly lower than those recorded by the sensors near high-traffic roads (Figure 2a). On the other hand, in the CAMx model, data is extracted from the nearest central mass point within a grid cell containing the AirNow station's location, providing an averaged representation of NO$_2$ levels within that specific grid cell. Consequently, the inherent positive bias in observations contributes to the model's tendency to underestimate this pollutant.

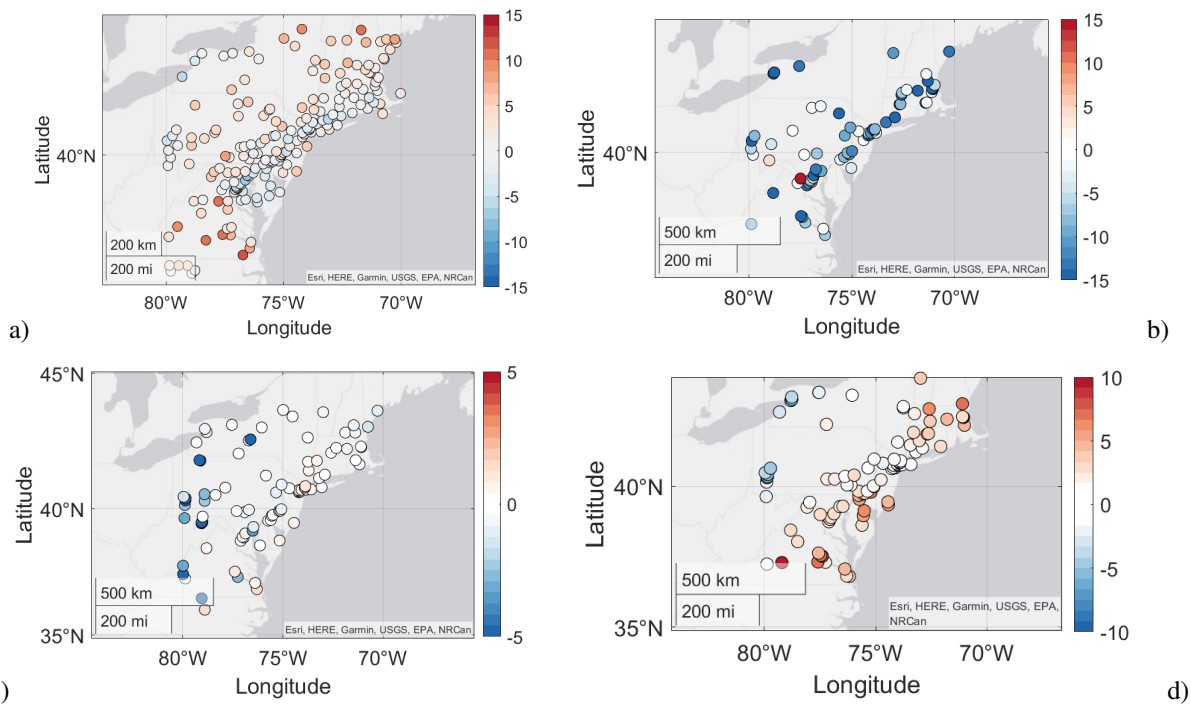

**Figure 4.** CAMx model performance against the AirNow observations; MBE calculated at each station for a) O$_3$ [ppb], b) NO$_2$ [ppb], c) SO$_2$ [ppb], and d) PM$_{2.5}$ [$\mu g/m^3$]. Blue shades show a systematic underestimation, while the red shades illustrate a systematic overestimation by the model.

## 4    Results and discussion

We study the impacts of ship emissions on the concentrations of four criteria pollutants: $PM_{2.5}$, $SO_2$, $NO_2$, and $O_3$. We calculate the time-average concentrations over different time periods depending on the pollutant as follows, to match the EPA national standards: one hour for $SO_2$ and $NO_2$; 8 hours for $O_3$; and 24 hours for $PM_{2.5}$. We analyze every pollutant from two perspectives: 1) from a regulatory perspective, thus calculating the statistics that are as close as possible to the EPA design value for each pollutant (Table 2), and 2) from a worst-case perspective, thus calculating the maximum contribution of ships to each pollutant over the entire three-month study period.

**Table 2.** Design values for criteria pollutants (U.S. Environmental Protection Agency (EPA), 2022b). For attainment purposes, the design values should be calculated by taking the average of the various percentiles over the past three years; since only one year was simulated in this study, the 3-year average could not be calculated and therefore only the actual percentiles were used.

| Pollutant | Design value | Threshold for attainment |
|---|---|---|
| $O_3$ | 4-th highest 8-hr averaged daily maximum | 70 ppb |
| $SO_2$ | 99-th percentile 1-hr daily maximum | 75 ppb |
| $NO_2$ | 98-th percentile 1-hr daily maximum | 100 ppb |
| $PM_{2.5}$ | 98-th percentile 24-hr average | $35\,\mu g\,m^{-3}$ |

To calculate the maximum contribution, we first find the differences between the two cases (WithShips minus NoShips) at every grid cell, averaged over the relevant time interval, which depends on the pollutant (Table 2); then, we find the maximum difference through the 3 months at every grid cell as follows:

$$\max(\Delta P_{i,j}) = \max_{t \in [1...n]} (P_{i,j}^{WithShips}(t) - P_{i,j}^{NoShips}(t)) \tag{1}$$

where $n$ is the number of data on the 1-, 8-, or 24-hr averaged pollutant P concentration values (the exact time averaging window depends on the pollutant, see Table 2) over the 3-month period of study, $P^{WithShips}$ and $P^{NoShips}$ are pollutant P concentrations with and without the ships, respectively, and $i$ and $j$ correspond to the model grid cell indices.

Although the maximum contribution from Eq. 1 is not valuable in terms of reaching or maintaining the EPA attainment for states, it is essential to understand the importance of maritime shipping on air quality, physically and statistically. A summary of the design values defined here for each pollutant to represent the EPA standards and the threshold for attainment are presented in Table 2. The defined design values follow the same criteria as defined by the EPA (U.S. Environmental Protection Agency (EPA), 2022b) but only for the time period of this study. For the remainder of the article, we will assume that the design values defined in Table 2 serve the purpose of analyzing the pollution from a regulatory perspective and are the same as the EPA standards for those pollutants for the time period of this study.

We find that the concentration of carbon monoxide (CO) remains unchanged in the presence of ships (not shown), suggesting that the CMV sector has minimal impact on the CO concentrations in the region. As such, we do not discuss the CO results in the rest of this study.

## 4.1 Fine particulate matter (PM$_{2.5}$)

The PM$_{2.5}$ species used in this study are those included in the CAMx model output (Ramboll Environment and Health, 2020). The EPA requires that the 3-year average of the 98-th percentile of the daily mean PM$_{2.5}$ concentrations should not exceed $35\,\mu g\,m^{-3}$. Here, we calculated the 98-th percentile of the 24-h averaged PM$_{2.5}$ concentrations at every grid cell during the simulation period in both scenarios, WithShips, and NoShips.

We find that PM$_{2.5}$ levels stayed below $35\,\mu g\,m^{-3}$ across most of the domain and that only two locations, i.e., Manhattan, NY, and Easton, PA (Figure 5a), crossed the $35\,\mu g\,m^{-3}$ maximum allowed concentrations and therefore were in non-attainment based on the design value defined in this study in Table 2. From a policy perspective, the CMV sector increases PM$_{2.5}$ levels up to $3.2\,\mu g\,m^{-3}$ in Manhattan, NY, and up to $2\,\mu g\,m^{-3}$ elsewhere (Figure 5c). This is while the percent contribution to PM$_{2.5}$ concentrations remains below 27% across the domain (Figure 5d). In a worst-case scenario, however, the maximum contribution of the ships to PM$_{2.5}$ concentrations within the 3 months is significantly high across the domain but due to the dominant southwesterly wind direction in the region (Golbazi et al., 2022), it mostly remains over the Atlantic Ocean (Figure 5b). The maximum impact on PM$_{2.5}$ during the 3 months reaches as high as $8\,\mu g\,m^{-3}$. Across the domain, the highest impacts are found offshore of MD and VA and in the Chesapeake Bay, DE. Over the land, the highest impacts are in Manhattan, NY, CT, and coastal MA.

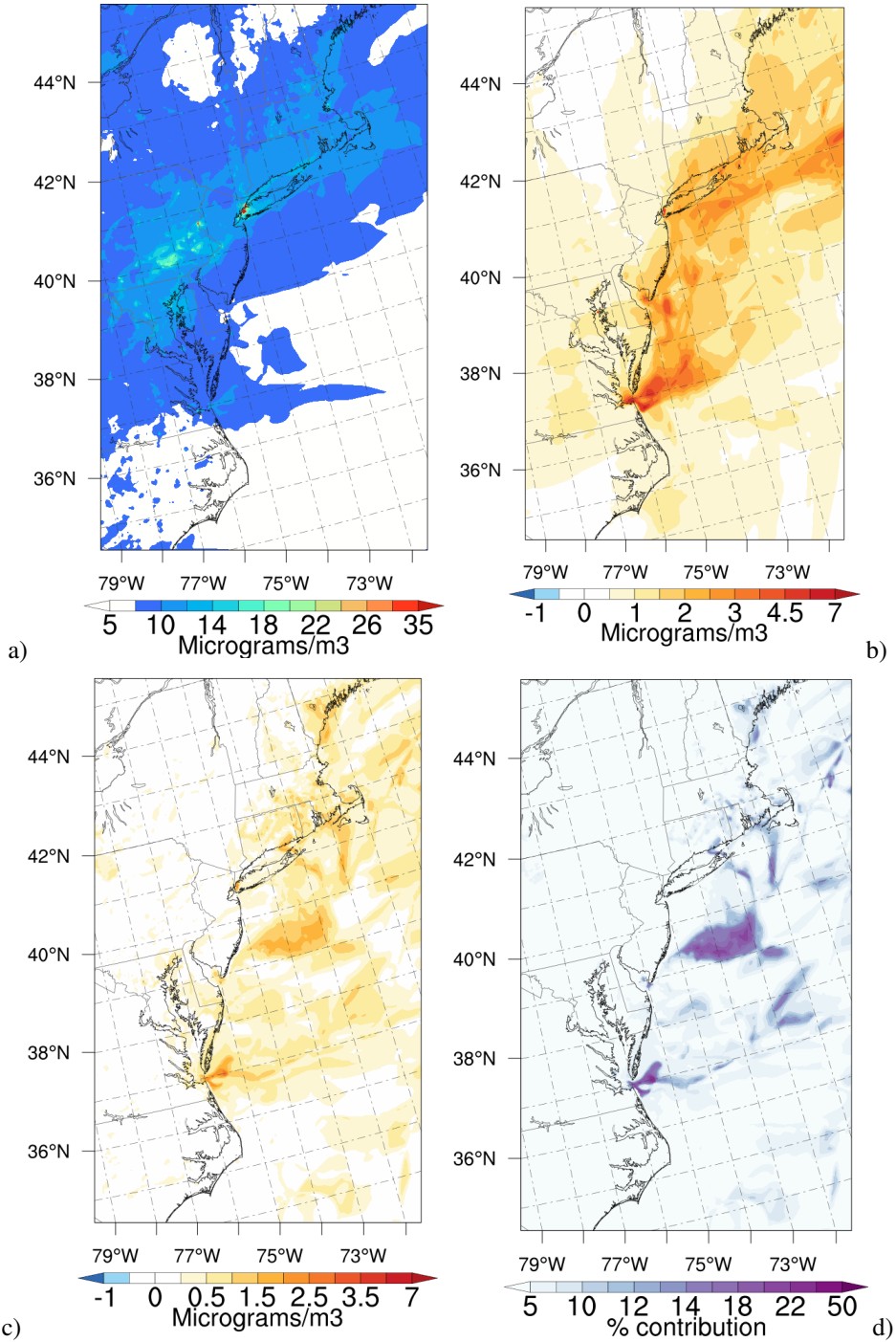

**Figure 5.** PM$_{2.5}$ concentration results: a) 98-th percentile of the 24-hr average PM$_{2.5}$ concentrations with ships (withShips); b) maximum contribution of the ships during the 3-month period (Eq. 1); c) difference between the two scenarios (WithShips minus NoShips) from a regulatory perspective, meaning the changes to the 98-th percentile PM$_{2.5}$ concentrations; and d) percent contribution of the ships to 98-th percentile of the 24-hr average PM$_{2.5}$ concentrations.

## 4.2  Sulfur dioxide (SO₂)

The $SO_2$ design value is defined as the 99-th percentile daily maximum $SO_2$ concentrations in the simulation period, which should not exceed 75 ppb (Table 2). Here we calculated the 99-th percentile of daily maximum $SO_2$ concentrations at every grid cell over the simulation period in the two scenarios, i.e., WithShips and NoShips. Then, we subtracted these two cases from one other (WithShips minus NoShips) to obtain the net effect of the maritime shipping sector.

Our results show that ships have a significantly high impact on $SO_2$ concentrations, up to 95% and 90% over the Atlantic Ocean and inland, respectively (Figure 6d). This suggests that the CMV sector is one of the highest contributors to $SO_2$ levels in the region. The increase in the $SO_2$ design value by ships remains mainly offshore and around the major shipping routes (Figure 6c). However, it reaches the interior of land in major ports and some parts of the coastal states. Over the simulation period, the contribution of the ships to the 99-th percentile daily $SO_2$ maxima is up to 45 ppb, with the highest impact in Baltimore, Maryland (MD), and Norfolk, Virginia (VA), and parts of New Jersey and Long Island (Figure 6c). We note that, however, the $SO_2$ design value in the region remained below 75 ppb in all states (Figure 6a). It is worth mentioning that the locations with the highest $SO_2$ concentrations are the ones highly impacted by the ships.

In addition, we calculated the 3-month maximum contributions of the ships to $SO_2$ concentrations, which indicates the worst-case scenario at every grid cell (figure 6b). Although the increase in $SO_2$ design value was mainly offshore, the maximum contribution of the ships to $SO_2$ showed a different pattern, with a maximum increase of up to ~185 ppb at a few grid cells around Norfolk, VA. We note that an occasional and short-term spike of high concentrations of $SO_2$, as we report here for Norfolk, is not necessarily associated with a strong health impact.

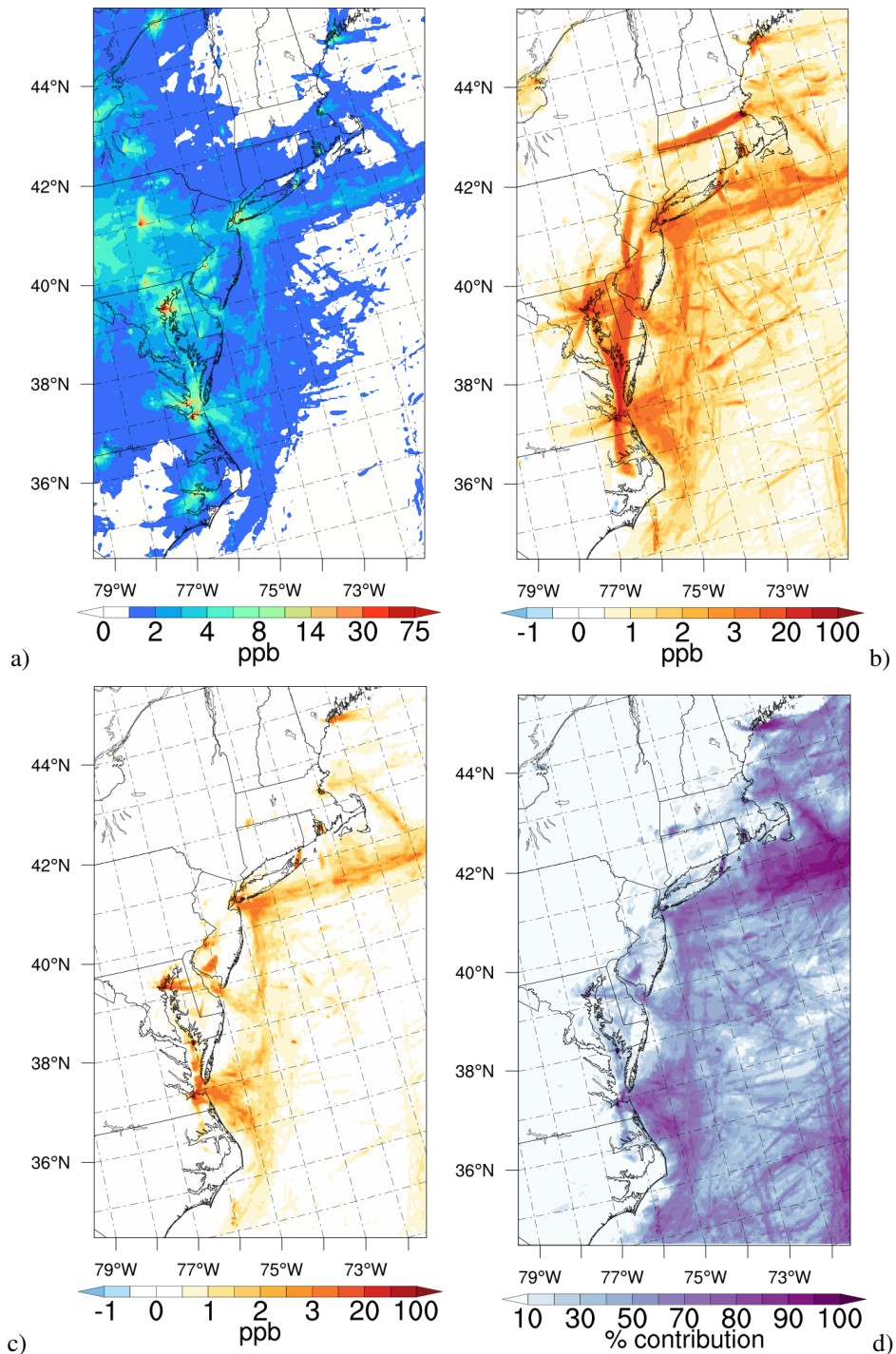

**Figure 6.** SO$_2$ concentration results: a) 99-th percentile of daily 1-hour maximum SO$_2$ concentrations with ships (WithShips); b) maximum contribution of the ships during the simulation period (Eq. 1); c) differences between the 99-th percentile daily maximum between the two scenarios (WithShips minus NoShips); and d) percent contribution of the ships to the 99-th percentile of the daily SO$_2$ maximum.

### 4.3 Nitrogen dioxide (NO$_2$)

NO$_2$ is a precursor to tropospheric ozone (O$_3$) formation, which has further negative impacts on human health (, EPA). NO$_2$ is generally directly emitted into the atmosphere from emission sources, including ships. We find that ships cause a significant increase in the 98-th percentile of daily maximum NO$_2$ concentrations, up to 34 ppb, but only at a few locations along the coast and in coastal states with major ports (Figure 7c), suggesting that, except for states with large ports, ships do not significantly impact the state compliance with the EPA standards. However, in NC, VA, DE, NY, and CT the shipping impacts reach beyond

15 ppb from a regulatory standpoint. Among these states, while NC remains in attainment with regulations (below 100 ppb), it experiences up to 80% contribution from the ships to its NO$_2$ concentrations. On the other hand, NY is the only state in the study domain that exceeds the 100 ppb standard for NO$_2$ concentrations, and shipping contribution to its non-attainment is 20-25%.

The maximum contribution of the ships to NO$_2$ concentrations, which is illustrated in Figure 7b, shows that, in a worst-case

scenario, ships contributed to up to ~50 ppb of NO$_2$ along the coast and 75 ppb over the Atlantic, which is significantly high compared to 100 ppb standard. The 3-month highest impact happens near the major ports and shipping routes but stretches to the land and over the ocean.

### 4.4 Ozone (O$_3$)

Tropospheric ozone is formed by both naturally occurring and anthropogenic sources. Ozone is not emitted directly into the air,

but, in the presence of sunlight, it is created by its precursors: VOC and NO$_x$. The rate of ozone production can be limited by either VOCs or NO$_x$. As a result, a specific location can be either VOC-limited or NO$_x$-limited. The rate of production/destruction of O$_3$ in the atmosphere is different in either of these regimes. We will further discuss this matter later in this section.

We use 8-hour average ozone concentrations for our analysis to maintain consistency with the EPA standards (U.S. Environmental Protection Agency (EPA), 2022b). We calculated the 8-hr averaged ozone values by averaging consecutive eight hours

of O$_3$ outputs at each hour of the day and storing it at the start hour (Cohen et al., 1999). For instance, O$_3$ at 11:00 in a day indicates the time-averaged O$_3$ concentrations between hours 11:00 and 19:00 in that day. Hereafter, we will refer to the 8-hr averaged O$_3$ concentrations simply as O$_3$ concentrations.

Ambient ozone concentrations are directly affected by temperature, solar radiation, wind speed, and other meteorological factors. Since O$_3$ production is a photochemical reaction, its peak concentrations are found during the daytime when tropospheric

ultraviolet radiation in the atmosphere is highest. Since the focus of this study is the daily high episodes of O$_3$ that are associated with adverse health impacts, we limit our analysis of the maximum impacts to only daytime hours. To select the daytime hours, we considered the 8-hr averaged O$_3$ daily profiles in 10 different locations along the coast from which 4 selected locations are shown in Figure 8. The lowest concentrations of O$_3$ are associated with hours 00:00 to 08:00 UTC (20:00 to 04:00 local time). We eliminated these hours from our analysis to only focus on concentrations during the high episodes. Therefore, we select the

295 hours with peak O$_3$ concentrations during the 24-h period i.e., 09:00 to 23:00 UTC (05:00 to 19:00 local time).

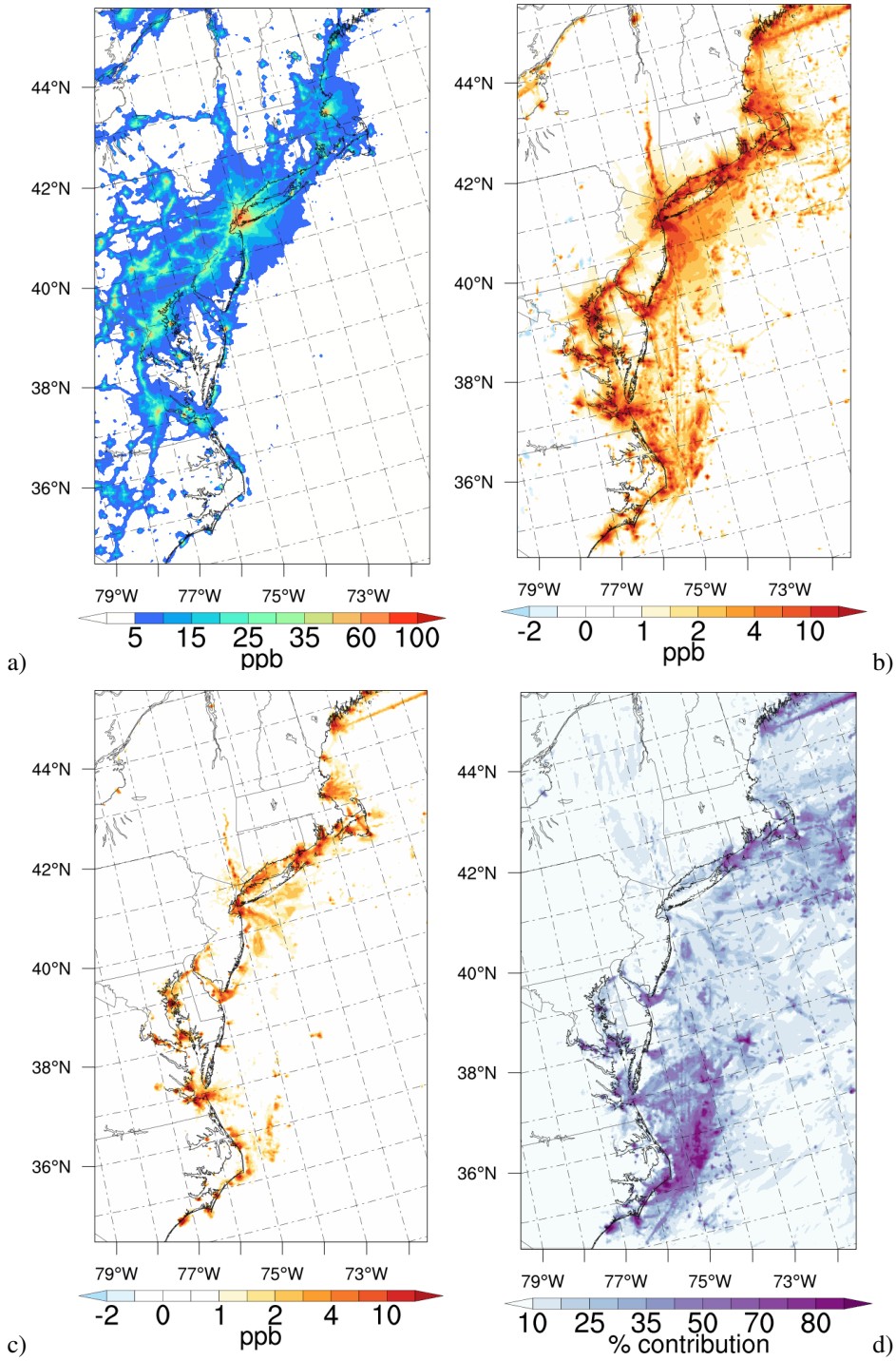

**Figure 7.** NO$_2$ concentration results: a) 98-th percentile of 1-hour daily maximum NO$_2$ concentrations with ships (WithShips); b) maximum contribution of the ships during the simulation period (Eq. 1); c) differences between the 98-th percentile of the 1-hour daily maximum between the two scenarios (WithShips minus NoShips); and d) percent contribution of the ships to the 98-th percentile of the daily NO$_2$ maximum.

We find that although ship emissions contribute to $O_3$ enhancement in the region, they reduce $O_3$ at some urban locations. We detect a significant increase or decrease in $O_3$ concentrations in the presence of the ships, depending on whether the location was $NO_x$ or VOC limited.

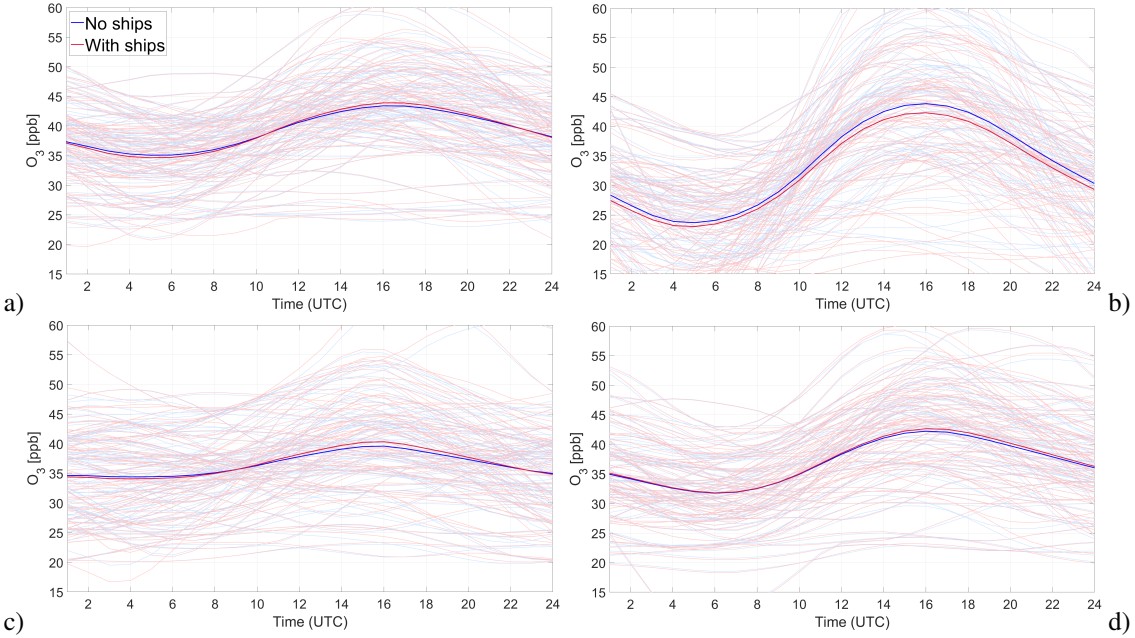

**Figure 8.** Eight-hour average $O_3$ concentration time series at four different locations along the East Coast from June $1^{st}$ to August $30^{st}$, 2018: a) Atlantic City, NJ; b) Manhattan, NY; c) Cape Cod, MA; and d) Providence, RI. The thin lines represent the daily time series and the thick lines are the 3-month average profiles with red lines for WithShips and blue lines for the NoShips simulations.

The maximum contribution of the ships to the $O_3$ levels over the entire 3-month period is illustrated in Figure 9c as a worst-case scenario. Over the ocean, the maximum increase is large, up to 8.6 ppb. However, the pollution increase remained primarily offshore and did not significantly impact the coastal areas. $O_3$ increased by 4-5 ppb at most in parts of North Carolina, Baltimore, MD, and parts of CT, and MA. Otherwise, the maximum increase over the land was up to 3.5 ppb. It's important to note that the maximum impact is not necessarily at the time when high $O_3$ episodes (from a regulatory perspective) are found. Despite the $O_3$ increase over the Atlantic, at the shore where the major ports are built, the maximum contribution of the ships is negative, suggesting that $O_3$ was destructed in the presence of the ships. This is due to the complex nature of atmospheric chemistry, where the fresh NO emissions from the ships scavenge $O_3$ and reduce its concentrations in urban VOC-limited areas.

While Figure 9b is important to understand the worst-case scenario of the shipping impact on $O_3$ pollution in the region, it does not help to measure the impact of these changes on state compliance with EPA standards. This is because a high impact on $O_3$ in the worst-case scenario may not correspond to the time of the day when $O_3$ daily maxima occurred. Therefore, to study the impacts of ship emissions from a policy perspective, it is necessary to explore the impacts on the $O_3$ design value at every grid cell. For $O_3$, the EPA defines this standard as the 4-th highest 8-hr averaged $O_3$ daily maxima, averaged over a 3-year period

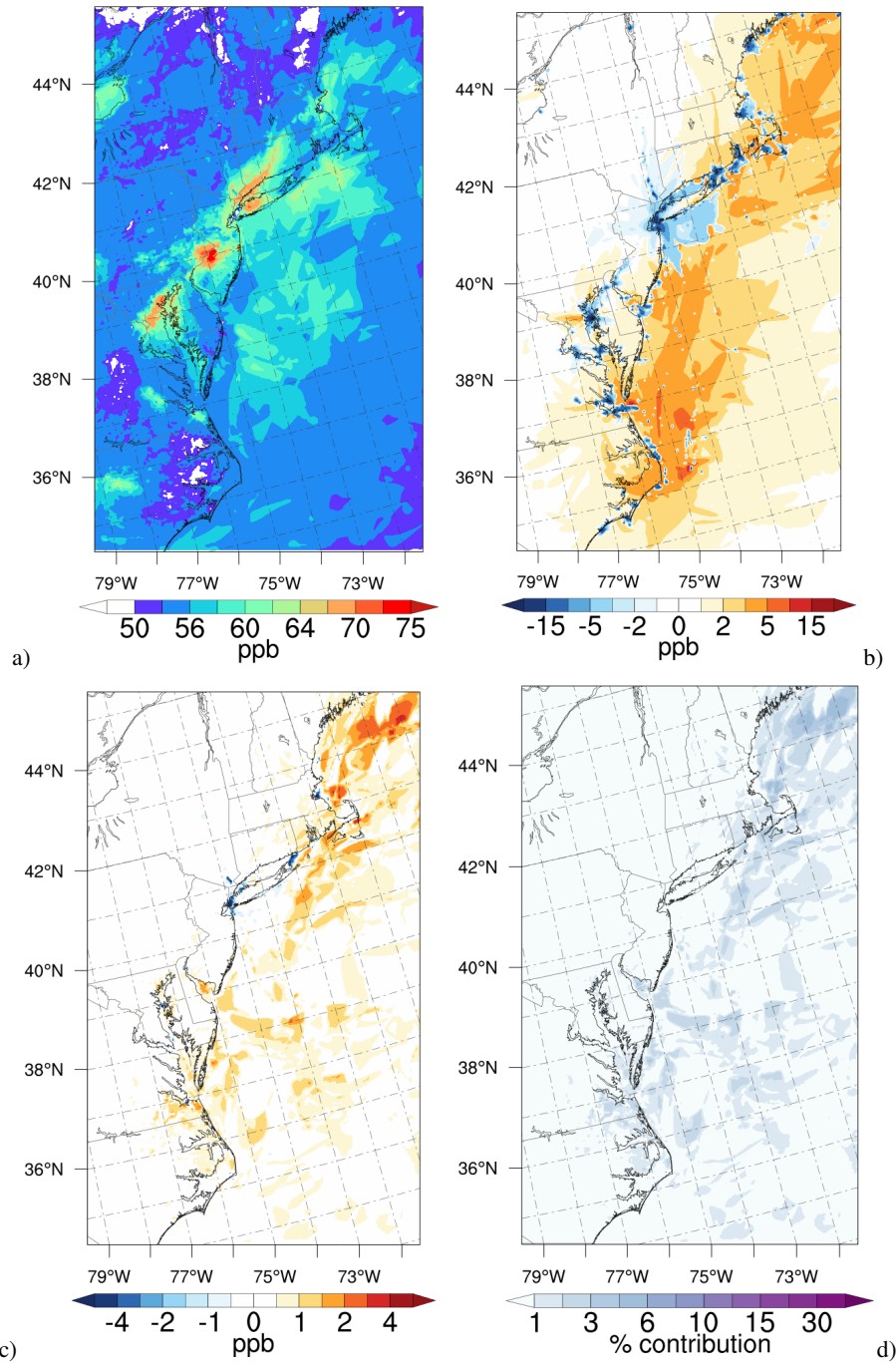

**Figure 9.** $O_3$ concentration results: a) 4-th highest 8-hour daily maximum $O_3$ concentrations with ships (WithShips); b) maximum contribution of the ships during the 3-month simulation period (Eq. 1); c) differences between the 4-th highest 8-hour daily maximum between the two scenarios (WithShips minus NoShips); and d) percent contribution of the ships to the 4-th highest daily $O_3$ maximum.

which should not exceed 70 ppb (U.S. Environmental Protection Agency (EPA), 2022b). Since we do not have data for three consecutive years, we focus on the 4-th highest daily maximum in our study period, the summer of 2018. Hereafter, we will refer to this value as $O_3$ design value in this study. We assume that this value represents the 4th highest daily maximum in the year 2018 since the highest $O_3$ episodes are expected to occur during the summer period. Thus, the regions with higher than 70 ppb $O_3$ concentrations in 9a are most susceptible to being in non-attainment with the EPA standards and therefore the impacts of the ships are of higher significance in those regions. Out of all states in the domain, NY, NJ, and MD are the only states that exceed the 70 ppb standard and are likely to be in non-attainment. All other states stay in attainment with the standards defined in Table 2 in either scenario.

From a policy perspective, $O_3$ design values increased in presence of the ships by up to 3.5 over the Atlantic Ocean. However, we find a reduction (up to 6.5 ppb) in $O_3$ concentrations at major ports along the East Coast (Figure 9c), where fresh NO is emitted by the ships into the atmosphere in VOC-limited regions (Figure 10). In most parts, the major impact of the ships remains offshore away from the coastal areas. However, in some regions in MA, RI, CT, ME, VA, NC, and MD ships contribute to $O_3$ increase at the coast from which, only MD is likely to be in non-attainment. The highest increase in $O_3$ design value inland is found in VA and NC and is up to 2.5 ppb, while we note that in NY, the 4-th highest daily maximum is decreased by 4 ppb in presence of the ships for the reasons discussed later in this section. However, the decrease in $O_3$ values remains in Manhattan, NY, and is not associated with the parts of Long Island (NY) where $O_3$ exceeds the standards.

In the atmosphere, the formation or destruction of ozone depends on the concentrations of both $NO_x$ and VOC and the ratio between them ($VOC/NOx$). Transportation usually is associated with high $NO_x$ emissions, therefore $O_3$ is generally $NO_x$-limited in rural areas and VOC-limited in urban areas, with low and high traffic densities, respectively.

In the VOC-limited regions, high concentrations of freshly emitted NO locally scavenge $O_3$ and lead to the formation of $NO_2$. Close to the emission sources, this titration process can be considered an ozone sink. In addition, high $NO_2$ concentrations deflect the initial oxidation step of VOC by forming other products such as nitric acid ($HNO_3$), which slows down the formation of $O_3$ (National Research Council, 1992; Beck et al., 1998). Because of these reactions, an increase in NO leads to a decrease in $O_3$ at VOC-limitted regions.

In CAMx, the VOC-limited regime is defined when the rate of change of hydrogen peroxide ($H_2O_2$) is lower compared to the rate of change of $HNO_3$. A higher $\Delta H_2O_2/\Delta HNO_3$ ratio indicates a $NO_x$-limited regime, while a lower $\Delta H_2O_2/\Delta HNO_3$ ratio corresponds to a VOC-limited regime. There are other indicators for determining the $NO_x$/VOC-limited regimes that are discussed in the literature (Li et al., 2022). However, here we use the $\Delta H_2O_2/\Delta HNO_3$ ratio as is used in the CAMx model (Ramboll Environment and Health, 2020).

To understand the formation/destruction of ozone in the presence of the ships in our study domain, we calculated the frequency of the VOC-limited regime based on the ratio at every grid cell. Figure 10a illustrates the percentage of the times that a VOC-limited regime occurred at every cell, which is the highest along the coast and in major cities. This indicates that $O_3$ may be affected by titration when ship emissions are present. We find that the NO concentrations increase along the coast where we detect a decrease in $O_3$ concentrations (Figure 10b) and a VOC-limited regime (Figure 10c).

This finding does not necessarily mean that ships help create better air quality since a reduction in $O_3$ is due to a significant increase in other important air pollutants i.e., $NO_x$ concentrations.

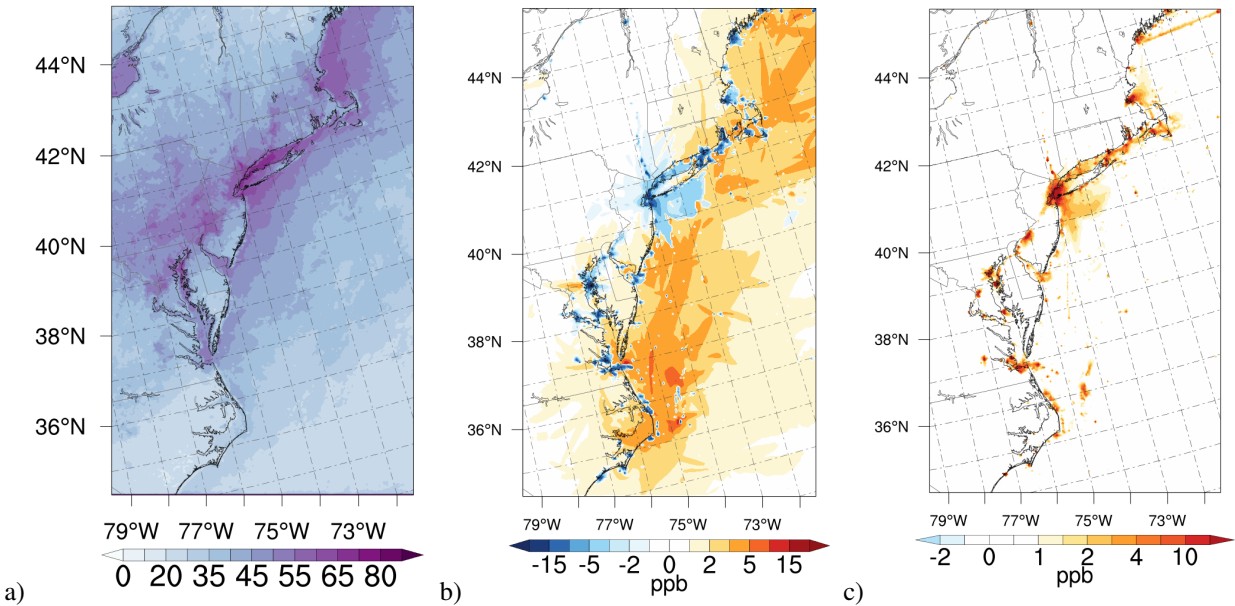

a)   b)   c)

**Figure 10.** VOC- versus $NO_x$-limited regime determination: a) percentage of the times when $\Delta H_2O_2/\Delta HNO_3 < 0.35$ as determined in CAMx model, which is an indication of a VOC-limited regime; b) maximum contribution of ships to $O_3$ pollution (Figure 9c); and c) same as in b) but for NO.

## 4.5 Diurnal Cycle of the impacts

In order to examine the diurnal variations in the impact of shipping activities on each of the four pollutants, we generated time series data representing the daily cycles of changes induced by ships. To achieve this, we specifically chose four key locations along the eastern coast: Manhattan, New York; Baltimore, Maryland; Boston, Massachusetts; and Norfolk, Virginia. This selection was deliberate, as these locations encompass large cities as well as major ports, making them suitable representatives for assessing the shipping-related effects on air quality.

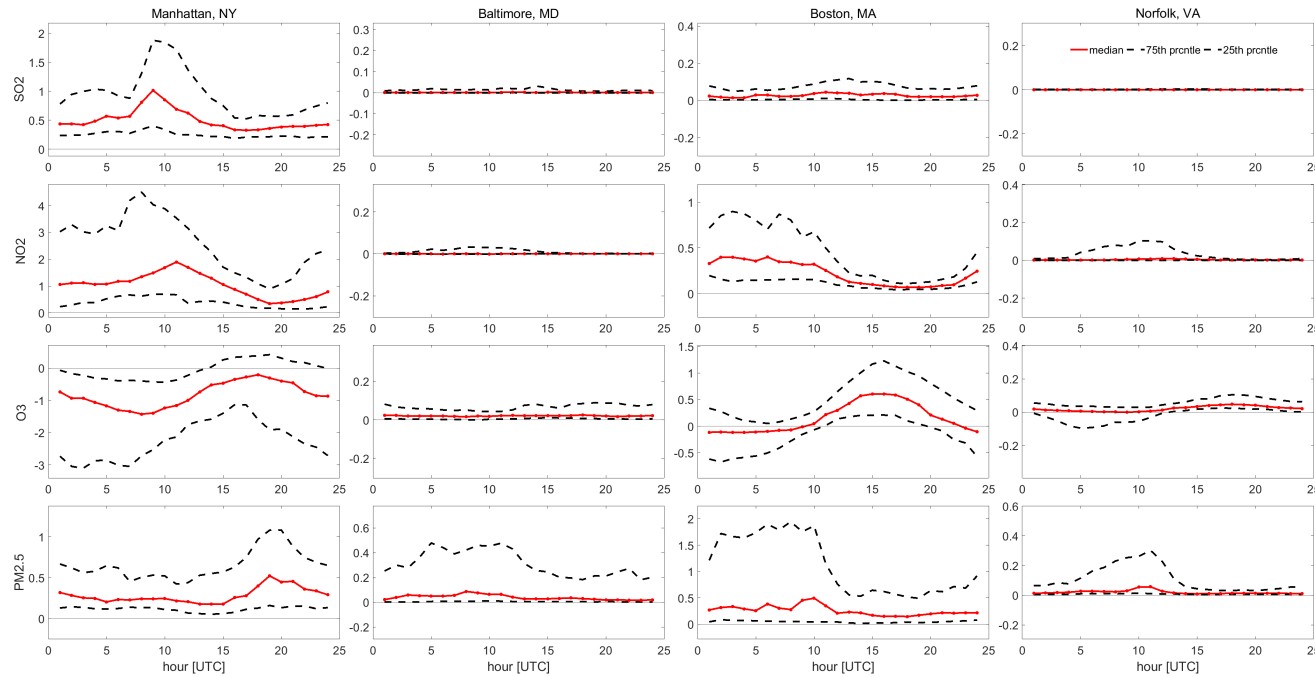

**Figure 11.** Diurnal cycle of the ship emission impacts on pollutants for a) $SO_2$ [ppb], b) $NO_2$ [ppb], c) $O_3$ [ppb], d) $PM_{2.5}$ [ $\mu g/m^3$ ]. The first through fourth columns represent changes in Manhattan, NY, Baltimore, MD, Boston, MA, and Norfolk, VA, respectively.

The dashed lines are the 25th and 75th percentiles, offering insights into the distribution of the impacts across various days
and stations at each simulation hour. In contrast, the solid pink line represents the median impact attributed to the presence of ships. For both $NO_2$ and $SO_2$, we observe an increase in concentrations when ships are present at all hours, as evidenced by the positive values of the median diurnal impact. Notably, the most significant impact for $NO_2$ is observed around 05:00 – 15:00 UTC (corresponding to 01:00 to 11:00 local time). For $SO_2$, we do not detect a clear diurnal pattern across all four locations. The median changes in $O_3$ levels show varying patterns across different locations. In Baltimore, MD, and Norfolk, VA, the
median impacts on $O_3$ are minimal. In Manhattan, NY, $O_3$ levels demonstrate consistent negative changes across all hours, indicating a reduction in $O_3$ concentration in the presence of ships, with the most pronounced decrease occurring between 05:00 - 12:00 UTC (equivalent to 01:00 - 08:00 local time). It's important to note that these values represent the 8-hour average $O_3$ concentrations, meaning that, for instance, 08:00 local time represents the average $O_3$ levels between hours 08:00 and 16:00. Conversely, in Boston, MA, the most significant impacts of ships on $O_3$ levels are observed between 11:00 and 20:00 UTC
(equivalent to 07:00 - 16:00 local time) and are increased.

PM$_{2.5}$ shows a similar diurnal pattern to $NO_2$ as it shows a positive impact (increase in PM$_{2.5}$ levels by the ships) in all hours, with the highest impact during the   00:00 – 12:00 UTC (corresponding to   20:00 to 08:00 local time). Apart from Manhattan, NY, where the highest impacts occur around hour 20:00 UTC (16:00 local time).

It's worth highlighting that the influence of shipping emissions on the four pollutants shown in Figures 5–9 (b-c) may be different than the findings in Figure 11. This divergence arises from our utilization of distinct metrics in these two analyses. In Figure 11, we base our assessment on median impacts within the four locations, whereas in the other figures, we evaluate the impacts with regard to EPA regulations or under a worst-case scenario.

## 5    Conclusions

Ships emit significant amounts of pollutants within 400 km of the shores. Here, we studied the ocean-going ship emissions on the air quality of the U.S. East Coast. We utilized the WRF-CAMx modeling system to simulate the pollution concentrations in the presence and absence of shipping activities along the East Coast and at the major ports. We used the WRF model to provide the meteorological inputs for the CAMx air quality model for the year 2018, on which the most recent EPA/NEI emission inventory is based. We particularly focused on $PM_{2.5}$, $SO_2$, $NO_2$, and $O_3$. Overall, we studied the outcomes of every pollutant from two perspectives: 1) from the EPA perspective concerning the national concentration standards for each specific pollutant, and 2) the maximum contribution of ships to that pollutant over the 3 months. Our assessment of the CAMx model's performance reveals strong performance in simulating $SO_2$ levels. The model shows a slight underestimation of $O_3$ concentrations near the coast and a slight overestimation farther from the shore. Nevertheless, the mean bias error for $O_3$ is limited to -1.12 ppb. Likewise, the bias in $PM_{2.5}$ concentrations remains below 5 $\mu g/m^3$. On the other hand, the model exhibits a noticeable underestimation of $NO_2$ concentrations, primarily stemming from a positive bias in observations collected in proximity to major roads.

We find that shipping increases the $PM_{2.5}$ concentrations across the domain. the 98-th percentile daily average $PM_{2.5}$ levels increased by 3.2 $\mu g\,m^{-3}$ over the ocean and in some coastal areas. However, in a worst-case scenario, ships contribute up to approximately 8.0 $\mu g\,m^{-3}$ to $PM_{2.5}$ concentrations, only over the Atlantic off the coast of MD, and VA. In addition, we find that ships have a significantly high impact, up to 95% and 90%, on the $SO_2$ concentrations over the Atlantic and inland, respectively. This suggests that the CMV sector is one of the highest contributors to $SO_2$ levels in the region. The shipping contribution to $SO_2$ levels was up to 45 ppb over coastal regions. Ship emissions also impacted the $NO_2$ design value by up to 34 ppb. In addition, our simulation results show that the impact of ship emissions on $O_3$ concentrations is not uniform, meaning that maritime shipping affects ozone pollution in both positive and negative ways. Although over the ocean $O_3$ concentrations increase significantly in the presence of ships (up to 8.6 ppb), in coastal areas with major cities and major ports $O_3$ concentrations decrease by up to ~6.5 ppb. To understand the reasons behind the $O_3$ reduction in the presence of ships, we analyzed the $\Delta H_2O_2/\Delta HNO_3$ ratio in the region, which is used to determine $NO_x$- or VOC-limited ozone production, as well as changes to NO concentrations, since they play a significant role in $O_3$ formation and destruction. We found that ships emit significant amounts of fresh NO in the atmosphere, which then helps scavenge $O_3$ in VOC-limited regimes. As a result, with higher NO concentrations in the atmosphere produced by ship emissions, $O_3$ is destructed in major cities and urban areas. By contrast, over the ocean (a $NO_x$-limited regime), excessive $NO_x$ emissions due to the ships contribute to the formation of $O_3$ and therefore an enhancement in $O_3$ concentrations. It is important to note that the destruction of $O_3$ by ship emissions in major cities does not necessarily mean that the ships create better air quality because a decrease in $O_3$ is a consequence of a significant increase in other pollutants

like NO. The diurnal cycle in the impact of shipping emissions across four major cities shows different patterns for different locations. For instance, the highest impacts on O3, occur at different times for different locations. $PM_{2.5}$ and $NO_2$, however, experience the highest changes in the early morning in most locations. On the other hand, we do not detect consistent patterns for changes in $SO_2$.

Overall, the majority of the time, due to the dominant southwesterly wind direction in the region, the impacts on different pollutants remained spatially confined offshore. However, in some coastal areas near the major ports, the impacts were significant.

## Data availability

The data for the model setup is available in the GitHub repository at:

https://github.com/golbazimaryam/ShippingEmissionsAndAirQuality

## Author contribution

M. Golbazi contributed to the design of the study, preparing the manuscript, setting up and carrying out simulations, and analysis. C.L. Archer contributed to the design of the project, preparing the manuscript and editing, as well as data analysis.

## Competing interests

"The authors declare that they have no conflict of interest."

*Acknowledgements.* *

Partial funding for this research came from the University of Delaware (UD) Graduate College Doctoral Fellowship and from the Delaware Natural Resources and Environmental Control (DNREC, award no. 18A00378). The simulations were conducted on the UD Caviness and NCAR Cheyenne high-performance computer clusters.

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
