# Peer review of "Impacts of maritime shipping on air pollution along the U.S. East Coast"

_Atmospheric Chemistry and Physics, 2023_

## Author Comment (AC1)

**Impacts of maritime shipping on air pollution along the U.S. East Coast**

**paper rebuttal:**

**Reviewer #1:**

This manuscript presented a modeling study of shipping emission impacts on air quality over east coast of the U.S. and reported significant impacts over some areas. Impacts of anthropogenic emissions on air pollution have been thoroughly investigated in past decades, yet the contribution from maritime shipping remains largely poorly documented. The study applied a solid modelling method with detailed discussions of the results and clear conclusions. In general the manuscript is well organized with good quality. Therefore, I would recommend the manuscript to be accepted with a minor revision, if the following comments could be properly addressed.

Main comment#1.

1. The manuscript lacks an evaluation section to demonstrate the performance of CAMx, which leaves the uncertainty remains unknown. It is necessary to show a discrepancy between simulation and observation so that the simulated contribution from shipping emissions can be properly interpreted.

**Author Response:**

We would like to thank the reviewer for their valuable suggestions. We believe that for differences between the two cases, the evaluation will not impact the results as much. That is because both scenarios are run with the same exact model setup and the only difference between the two sets of runs, is the presence and absence of the ships. For that reason, if a pollutant is under/overestimated in one set of runs with the ships, it is also under/overestimated in the other set of runs without the ships. However, due to the non-linear and complex nature of the atmosphere, and to maintain transparency in our research and with respect to the reviewer's comment, we have completed the analysis on model performance evaluation and have added it as a section (CAMx model performance analysis) to our paper as follows:

> *"CAMx model performance analysis*
>
> The primary goal of this study is to explore changes to pollution levels between the two examined case studies, one involving the presence of ships and the other without. Despite the instances where CAMx may either under or overestimate pollutant concentrations, it is noteworthy that the model bias remains the same in both scenarios. Consequently, we hold the view that these outcomes are unlikely to have a significant influence on our analysis. Nevertheless, we have thoroughly evaluated the model's performance to maintain transparency in our findings. It's important to acknowledge that uncertainties in air quality modeling can arise from various sources, such as uncertainties in emission inventories (Foley et al., 2015), the accuracy of meteorological inputs (Kumar et al., 2019; Ryu et al., 2018; Zhang et al., 2007), numerical noise inherent in the model (Ancell et al., 2018; Golbazi et al., 2022), and numerical approximations.

For our evaluation process of these four pollutants, we rely on measurement data sourced from the Environmental Protection Agency (EPA) AirNow program, which is publicly accessible (https://aqs.epa.gov/aqsweb/documents/data_api.html ). Within the geographical scope of our study, we have access to data from a network of monitoring stations. Specifically, there are a total of 196 stations providing data for ozone (O3), and 87, 73, and 118 stations supplying data for SO2, NO2, and PM2.5, respectively. This extensive dataset forms the basis of our assessment, enabling us to comprehensively evaluate the CAMx model's performance in replicating real-world air quality conditions for these pollutants. It is worth mentioning that evaluating PM2.5 presents challenges due to the nature of EPA-reported PM2.5 measurements in the AirNow database. These values are directly obtained through instrumental measurements, classifying any particle smaller than 2.5 micrograms as a PM2.5 species. This method doesn't provide a clear means of distinguishing between the various particles detected by these instruments. In contrast, the PM2.5 species in our study are defined based on CAMx model documentation (Rambol, 2020). This divergence in approach makes a comprehensive PM2.5 evaluation challenging and pursuing alternative assessment methods falls beyond the scope of our current study.

[Figure]

*Figure 3., Model bias time series; CAMx model performance evaluated against the AirNow measurements; MBE is calculated across all stations at each day a) O3 [ppb], b) NO2 [ppb], and c) SO2 [ppb], and d) PM2.5 [µg/m3].*

Figure 3 illustrates a time series of the AirNow measurements across the simulation days (in black circles), as well as the co-located CAMx outputs for the pollutant of interest in the solid black line. The co-located data are such that they are extracted at the same hour as observations and at the mass point of the grid cell that contains that specific station. Figure 4, on the other hand, illustrates the MBE calculated at every station and depicts a spatial distribution of the model MBE for each pollutant using the co-located data.

CAMx demonstrates a tendency to slightly under- or over-estimate O3 concentrations closer to the coast, and away from the coast, respectively (Figure 2a). Our focus is mainly on locations closer to the coast since that is where we detect the highest impact of shipping emissions. For O3, a calculated mean bias error (MBE) of -1.12 ppb indicates a systematic underestimation of around 2.5% across all monitoring stations within the designated domain. Overall, the model effectively captures the O3 trend and demonstrates a satisfactory level of agreement with observational data, as illustrated in Figure 1a.

[Figure]

*Figure 4. CAMx model performance against the AirNow observations; MBE calculated at each station for a) O3 [ppb], b) NO2 [ppb], and c) SO2 [ppb], and d) PM2.5 [µg/m3]. Blue shades show a systematic underestimation, while the red shades illustrate a systematic overestimation by the model.*

In addition, CAMx showcases a strong alignment with observational data in terms of SO2 simulations with minimal deviation from the observations. For PM2.5, the model typically underestimates high PM2.5 episodes, as is commonly observed in prior studies (Delle, 2020; Golbazi, 2023). Nonetheless, for the remainder of the time, it demonstrates a strong alignment with observed data, as shown in Figure 3d. Figure 2d reveals that the model bias for PM2.5 consistently remains below 5 µg/m3 for the majority of coastal stations, with only a few exceptions. Shifting focus to NO2, the model systematically underestimates NO2 concentrations (Figure 3b, and Figure 4b). This observation aligns with findings reported in existing literature (Ma et al., 2006). The notable underestimation of NO2 levels within the

model can be attributed to the fact that the monitoring stations are typically situated in close proximity to major roadways characterized by heavy traffic flow, resulting in elevated NO2 emissions. Conversely, NO2 concentrations at locations farther away from these monitoring stations tend to be significantly lower than those recorded by the sensors near high-traffic roads (Figure 2a). On the other hand, in the CAMx model, data is extracted from the nearest central mass point within a grid cell containing the AirNow station's location, providing an averaged representation of NO2 levels within that specific grid cell. Consequently, the inherent positive bias in observations contributes to the model's tendency to underestimate this pollutant."

Main comment#2.

2. Another issue with the manuscript is the lack of necessary discussion about the diurnal variational and seasonality of shipping emissions on coastal air quality. Wind patterns differ significantly between day- and night-time and this may play an important role in coastal cities air quality as it affects both chemistry and dispersion. For example, it may alter the NOx and VOC relative ratio, and bring excessive chlorine from the ocean to urban. Fig.5 showed a diurnal pattern of O3, but no in-depth discussion was given to explain the diurnal changes. For the same reason, the contribution from shipping emissions may be also different between summer and winter. Therefore, I would recommend the manuscript to include at least a brief discussion regarding this seasonality according to other studies, if adding a winter simulation was not feasible.

**Author Response:**

We agree with the reviewer that ships will have different impacts during different seasons. However, our main focus in this study is O3 pollution on the East Coast which is directly affected by temperature, solar radiation, wind speed, and other meteorological factors. For this reason, we base our study on summer when the highest O3 episodes occur. In addition, the emission inventory from the EPA was limited to April – October of 2018 and it was not possible to extend our study to create winter simulations. We understand that a seasonality analysis of the ship impacts would be a valuable addition, however, as the reviewer has noted, due to the computational costs of the simulations and limitations on the emissions, it is not feasible at this time to repeat our simulations for other seasons. To cover this topic, however, we explored this matter in the literature. To the best of our knowledge, there aren't many studies exploring shipping pollution in the United States. However, we found fundamental studies that are conducted for other regions in the world, or on a global scale and have added the paragraph below about their findings for the seasonal variations of the shipping impacts. To organize the order of the topics with the new addition to the "Introduction" section, we have re-ordered the last few paragraphs in the "Introduction" after adding the new studies. The new addition to our introduction reads as follows in the revised paper:

"Seasonal variations in the impact of shipping on various pollutants have been documented in prior studies. For example, Eyring et al. (2010) noted that during Mediterranean summer conditions, characterized by slow atmospheric transport, strong solar radiation, and limited washout, primary ship emissions accumulate, and secondary pollutants form. They reported that secondary sulfate aerosols from shipping were responsible for 54% of the average sulfate aerosol concentration in the region during the summer. Our findings along the US East Coast

align with their finding, highlighting the substantial contribution of ships to SO2 pollution during the summer season.

Furthermore, they observed that in winter, shipping NOx emissions could lead to ozone depletion in northern Europe. In a separate study, Eyring et al., (2007) noted significant variations in simulated O3 levels between January and July, despite a consistent ship emission inventory throughout the year. They found that during winter, additional NOx emissions from shipping led to O3 reduction due to titration, while in summer, these emissions resulted in relatively modest but positive O3 concentration changes in regions with sufficient solar radiation. In addition, they show that the highest ship impacts on O3 due to the ship emissions were found in July and April, whereas in October and January, the impacts were significantly smaller."

In response to the reviewer's feedback concerning the analysis of diurnal changes in the impact of shipping emissions on the pollutants, we generated a time series representing these impacts. Constructing a diurnal data series required employing statistical methods, including averaging data across the other two dimensions out of the three (averaging over days and locations).

However, it's crucial to note that averaging ship impact data across all grid cells would obscure the true impact, as ships primarily affect specific grid cells near the coast, and including unaffected grid cells would impact our statistics. To avoid this issue, we strategically selected four specific coastal locations, each corresponding to a significant city or major port. For each pollutant, we extracted the time series data from these chosen locations (as shown in Figure 11 in the new revision). This selection was made to ensure the representation of large urban areas and key ports in our analysis. Below, we provide additional context including our findings regarding the diurnal impacts.

"In order to examine the diurnal variations in the impact of shipping activities on each of the four pollutants, we generated time series data representing the daily cycles of changes induced by ships. To achieve this, we specifically chose four key locations along the eastern coast: Manhattan, New York; Baltimore, Maryland; Boston, Massachusetts; and Norfolk, Virginia. This selection was deliberate, as these locations encompass large cities as well as major ports, making them suitable representatives for assessing the shipping-related effects on air quality.

[Figure]

*Figure 11. Diurnal cycle of the ship emission impacts on pollutants for a) SO2 [ppb], b) NO2 [ppb], c) O3 [ppb], d) PM2.5 [ µg/m3]. The first through fourth columns represent changes in Manhattan, NY, Baltimore, MD, Boston, MA, and Norfolk, VA, respectively.*

The dashed lines are the 25th and 75th percentiles, offering insights into the distribution of the impacts across various days and stations at each simulation hour. In contrast, the solid pink line represents the median impact attributed to the presence of ships.

For both NO2 and SO2, we observe an increase in concentrations when ships are present at all hours, as evidenced by the positive values of the median diurnal impact. Notably, the most significant impact for NO2 is observed around 05:00 – 15:00 UTC (corresponding to 01:00 to 11:00 local time). For SO2, we do not detect a clear diurnal pattern across all four locations. The median changes in O3 levels show varying patterns across different locations. In Baltimore, MD, and Norfolk, VA, the median impacts on O3 are minimal. In Manhattan, NY, O3 levels demonstrate consistent negative changes across all hours, indicating a reduction in O3 concentration in the presence of ships, with the most pronounced decrease occurring between 05:00 - 12:00 UTC (equivalent to 01:00 - 08:00 local time). It's important to note that these values represent the 8-hour average O3 concentrations, meaning that, for instance, 08:00 local time represents the average O3 levels between hours 08:00 and 16:00. Conversely, in Boston, MA, the most significant impacts of ships on O3 levels are observed between 11:00 and 20:00 UTC (equivalent to 07:00 - 16:00 local time) and are increased. PM2.5 shows a similar diurnal pattern to NO2 as it shows a positive impact (increase in PM2.5 levels by the ships) in all hours, with the highest impact during the ~ 00:00 – 12:00 UTC (corresponding to ~ 20:00 to 08:00 local time). Apart from Manhattan, NY, where the highest impacts occur around hour 20:00 UTC (16:00 local time).

It is worth highlighting that the influence of shipping emissions on the four pollutants shown in Figures 6–9 (b-c) may be different than the findings in Figure 11. This divergence arises from our utilization of distinct metrics in these two analyses. In Figure 11, we base our assessment on median

impacts within the four locations, whereas in the other figures, we evaluate the impacts with regard to EPA regulations or under a worst-case scenario.

Main comment#3.

3. Another main issue with the manuscript is lack of thorough discussion of biomass burning impact on precipitation. First, biomass burning intensity was not clearly described. Fig.7 showed fire spot on Nov.25, but the study period is Dec.14-19, the gap is too long to indicate the rapid changes of both fire and precipitation. There is no demonstration about fire emission 2~3 days prior to the study period, assuming the extreme precipitation may turn off the local biomass burning. Second, to distinguish the impact of biomass burning aerosol, it is necessary to explain the influence of using diagnostic aerosol first. This would help to understand the relative contribution from anthropogenic aerosol and biomass burning emissions.

**Author Response:**

It seems like this comment is not relevant to our study. The comment is for Figure 7 which shows a fire spot for Nov 25. The reviewer mentions that our analysis is for Dec 14-17.

Our study spans over 3 months of summer in 2018, and we do not present any results for Nov 25. In addition, Figure 7 in our study shows the "a) percentage of the times when $\Delta H2O2/\Delta HNO3 < 0.35$ as determined in the CAMx model, which is an indication of a VOC-limited regime; b) maximum contribution of ships to O3 pollution (Figure 6c); and c) same as in b) but for NO".

Minor comment#1.

4. It's necessary to describe the vertical layer configuration (e.g., first layer height) to demonstrate that the near-surface layers can properly resolve the stack heights of shipping emissions.

**Author Response:**

We appreciate the reviewer's concern on this important matter. We have corrected the statement below to include our model configuration in vertical as well as in horizontal directions:

"The meteorological files have 400 x 400 horizontal grid points covering the entire CAMx domain, which consists of 315 x 300 grid points, the same as the emission files. We impose 35 vertical levels that are closely spaced near the surface and then gradually expand. The top hydrostatic pressure is 20 hPa and the lowest model level is at approximately 3.5 m above mean sea level (AMSL). Details about the model configuration are discussed in Table 1."

**NOTE:** We updated the conclusions as needed for the updates as follows:

"Ships emit significant amounts of pollutants within 400 km of the shores. Here, we studied the ocean-going ship emissions on the air quality of the U.S. East Coast. We utilized the WRF-CAMx modeling system to simulate the pollution concentrations in the presence and absence of shipping activities along the East Coast and at the major ports. We used the WRF model to provide the

meteorological inputs for the CAMx air quality model for the year 2018, on which the most recent EPA/NEI emission inventory is based. We particularly focused on PM2.5, SO2, NO2, and O3.

Overall, we studied the outcomes of every pollutant from two perspectives: 1) from the EPA perspective concerning the national concentration standards for each specific pollutant, and 2) the maximum contribution of ships to that pollutant over the 3 months. Our assessment of the CAMx model's performance reveals strong performance in simulating SO2 levels. The model shows a slight underestimation of O3 concentrations near the coast and a slight overestimation farther from the shore. Nevertheless, the mean bias error for O3 is limited to -1.12 ppb. Likewise, the bias in PM2.5 concentrations remains below 5 µg/m3. On the other hand, the model exhibits a noticeable underestimation of NO2 concentrations, primarily stemming from a positive bias in observations collected in proximity to major roads.

We find that shipping increases the PM2.5 concentrations across the domain. the 98-th percentile daily average PM2.5 levels increased by 3.2 µg/m3 over the ocean and in some coastal areas. However, in a worst-case scenario, ships contribute up to approximately 8.0 µg/m3 to PM2.5 concentrations, only over the Atlantic off the coast of MD, and VA. In addition, we find that ships have a significantly high impact, up to 95% and 90%, on the SO2 concentrations over the Atlantic and inland, respectively. This suggests that the CMV sector is one of the highest contributors to SO2 levels in the region. The shipping contribution to SO2 levels was up to 45 ppb over coastal regions. Ship emissions also impacted the NO2 design value by up to 34 ppb. In addition, our simulation results show that the impact of ship emissions on O3 concentrations is not uniform, meaning that maritime shipping affects ozone pollution in both positive and negative ways. Although over the ocean O3 concentrations increase significantly in the presence of ships (up to 8.6 ppb), in coastal areas with major cities and major ports O3 concentrations decrease by up to ~6.5 ppb. To understand the reasons behind the O3 reduction in the presence of ships, we analyzed the $\Delta H2O2/\Delta HNO3$ ratio in the region, which is used to determine NOx- or VOC-limited ozone production, as well as changes to NO concentrations, since they play a significant role in O3 formation and destruction. We found that ships emit significant amounts of fresh NO in the atmosphere, which then helps scavenge O3 in VOC-limited regimes. As a result, with higher NO concentrations in the atmosphere produced by ship emissions, O3 is destructed in major cities and urban areas. By contrast, over the ocean (a NOx-limited regime), excessive NOx emissions due to the ships contribute to the formation of O3 and therefore an enhancement in O3 concentrations. It is important to note that the destruction of O3 by ship emissions in major cities does not necessarily mean that the ships create better air quality because a decrease in O3 is a consequence of a significant increase in other pollutants, like NO. The diurnal cycle in the impact of shipping emissions across four major cities shows different patterns for different locations. For instance, the highest impacts on O3, occur at different times for different locations. PM2.5 and NO2, however, experience the highest changes in early morning in most locations. On the other hand, we do not detect consistent patterns for changes in SO2.

Overall, the majority of the time, due to the dominant southwesterly wind direction in the region, the impacts on different pollutants remained spatially confined offshore. However, in some coastal areas near the major ports, the impacts were significant."

---

## Author Comment (AC3)

**Impacts of maritime shipping on air pollution along the U.S. East Coast**

**paper rebuttal:**

**Reviewer #2:**

This manuscript offers an assessment of the effects of shipping emissions on air quality along the eastern coast of the United States pointing to substantial impacts within certain regions. I found the discussion to be well-reasoned and insightful, especially when tackling the dynamics between nitrogen oxides, ozone, and VOCs. Also, the perspective added by the authors on meeting the emission regulations currently in place in the U.S. is quite relevant. Thus, with a minor revision addressing the subsequent remarks, I am inclined to recommend its acceptance.

These include a more detailed introduction of the emission inventories (e.g., what anthropogenic sources have been included? And what quantity?), so the reader can better understand the share of shipping within the region studied. A map with the spatial representation of those emission inventories would also be relevant to understanding how the concentrations change compared to where emissions occur. In addition, if authors are willing to discuss the impact of shipping in different U.S. states, it would be relevant to include their location and name in at least one figure to contextualize the non-American readership.

**Author Response:**

Thanks for bringing this point up, we have added names of some important locations to the map of the domain for unfamiliar readers.

[Figure]

The reviewer brings up a valuable point that showing a map distribution of emissions would help understand the changes in concentrations compared to the emissions. However, an illustration of the spatial distribution of all emissions is challenging, in that, the emissions are provided in different formats (2D emissions and elevated point sources). In addition, in the 2D gridded emissions, several pollutants are included which are primary pollutants. With respect to the reviewer's comment, we produced maps of the 2D gridded emissions. But out of the four pollutants that we study here, only two (NO2 and SO2) are directly emitted into the atmosphere and therefore are included in emission data. We provided the plots for those two pollutants and added them to the "Emission Data" section with the description below:

[Figure]

Figure 2. Gridded 2D emission distribution across the domain (averaged over three months) in moles/s for a) NO2, and b) SO2. The gridded emissions include all the 2D anthropogenic and biogenic emissions and exclude the elevated point sources.

"The spatial distribution of the 2D gridded merged anthropogenic emissions are illustrated in Figure 2. It's important to note that O3 is a secondary pollutant, meaning it isn't directly emitted into the atmosphere. Conversely, PM2.5 is either a primary or secondary pollutant. Hence, we have specifically generated gridded emission maps for NO2 and SO2, only. The distribution of NO2 emissions closely mirrors the pattern of major highways and roads, as transportation stands out as one of the most significant sources of nitrogen oxide (NOx) emissions. The objective of this figure is to explain the spatial distribution of gridded anthropogenic emissions, shedding light on how concentrations change (Figures 6a and 7a) in relation to their emission sources."

About the emission inventory, we originally included a subsection (Emission Data) under the Methods section. However, we realized that it is short and may not be clear. In the original submission, we had:

"*In the 2018 NEI data, the gridded 2D emissions are merged, meaning that they are provided as one set of surface emissions that include all emission sectors. On the other hand, the elevated point sources are provided for each potential sector separately. For ship emissions, we use the emission data for the Commercial Marine Vessels (CMV) sector, which includes Category 1, 2 (small engine), and 3 (large engine) ships. CAMx computes the time-varying buoyant plume rise using stack parameters and the hourly emissions for each emissions sector, including CMV. Unlike previous EPA data sets, the CMV emissions in 2018 are at a one-hour temporal resolution. The initial and boundary conditions for this study are also provided by the EPA.*"

To address the reviewer's concern, we have now expanded this paragraph about the emission inventory and different sectors within it as follows:

"In the 2018 NEI data is based on the year 2017 activity. It contains merged gridded 2D surface emissions, meaning that they are provided as one set of surface emissions that include all the existing 2D emission sectors, i.e., anthropogenic emissions including aircraft emissions, on-road and non-road emissions, railroad emissions, and agricultural emissions. It also includes biogenic emissions. The 2018 inventory, however, lacks the wildfire emissions for this time and domain. Our investigation through the wildfire history showed that 2018 was a year with a low number of wildfires especially along the East Coast (https://www.nifc.gov/fire-information/statistics/wildfires) and therefore we do not believe this to significantly impact our findings. However, in future studies, including wildfire emissions upon availability is recommended. In contrast to the 2D grided emissions, the elevated point sources in this inventory are provided for each sector, separately.

For ship emissions, we use the emission data for the Commercial Marine Vessels (CMV) sector, which includes Category 1, 2 (small engine), and 3 (large engine) ships. These emissions are calculated based on the ship's fuel consumption, ship engine type, ship activity, and emission factors specific to those characteristics. EPA's CMV estimates are computed using detailed satellite-based automatic identification system (AIS) activity data from the US Coast Guard (EPA, 2017 & 2020). Unlike previous EPA data sets, the CMV emissions in 2018 are at a one-hour temporal resolution, which is very important and makes this study the first to utilize hourly emissions for the ships.

Other point sources present in this inventory include electric generation units, point oil, and gas sources, and any other point sources. CAMx computes the time-varying buoyant plume rise using stack parameters and the hourly emissions for each emissions sector, including CMV. The initial and boundary conditions for this study are also provided by the EPA and are products of the GEOS-Chem model."